# Modified Carbon Nanotubes: Surface Properties and Activity in Oxygen Reduction Reaction

**Vera Bogdanovskaya *** , **Inna Vernigor, Marina Radina, Vladimir Sobolev, Vladimir Andreev** and **Nadezhda Nikolskaya**

A.N. Frumkin Institute of Physical Chemistry and Electrochemistry, Russian Academy of Sciences, 119071 Moscow, Russia; msnoviinna@gmail.com (I.V.); merenkovamarina@mail.ru (M.R.); vsobolev@phyche.ac.ru (V.S.); vandr@phyche.ac.ru (V.A.); nfn1910-55@yandex.ru (N.N.)
* Correspondence: bogd@elchem.ac.ru

**Abstract:** In order to develop highly efficient and stable catalysts for oxygen reduction reaction (ORR) that do not contain precious metals, it is necessary to modify carbon nanotubes (CNT) and define the effect of the modification on their activity in the ORR. In this work, the modification of CNTs included functionalization by treatment in NaOH or $HNO_3$ (soft and hard conditions, respectively) and subsequent doping with nitrogen (melamine was used as a precursor). The main parameters that determine the efficiency of modified CNT in ORR are composition and surface area (XPS, BET), hydrophilic–hydrophobic surface properties (method of standard contact porosimetry (MSP)) and zeta potential (dynamic light scattering method). The activity of CNT in ORR was assessed following half-wave potential, current density within kinetic potential range and the electrochemically active surface area ($S_{EAS}$). The obtained results show that the modification of CNT with oxygen-containing groups leads to an increase in hydrophilicity and, consequently, $S_{EAS}$, as well as the total (overall) current. Subsequent doping with nitrogen ensures further increase in $S_{EAS}$, higher zeta potential and specific activity in ORR, reflected in the shift of the half-wave potential by 150 mV for $CNT_{NaOH-N}$ and 110 mV for $CNT_{HNO3-N}$ relative to $CNT_{NaOH}$ and $CNT_{HNO3}$, respectively. Moreover, the introduction of N into the structure of $CNT_{HNO3}$ increases their corrosion stability.

**Keywords:** carbon nanotubes; modification; oxygen electrochemical reduction; electrochemically active surface area; corrosion resistance





## 1. Introduction

Carbon nanotubes (CNT) are one of the most studied carbon materials (CM) for electrocatalysts [1]. They are characterised by high strength, large specific surface area, stability in alkaline and acidic media. However, much attention is paid to changing their electronic properties in order to increase chemical stability and catalytic activity [2]. Electronic properties can be tailored by modifying the CNT surface, including the formation of oxygen-containing functional groups [3] and/or introducing heteroatoms into the CNT structure [4]. Treating CNT with strong acids or alkalis leads to the formation of oxygen functional groups on the surface, such as carbonyl, carboxyl, hydroxyl, ketone and alcohol, enhancing electrocatalytic properties of CNT [5,6]. In works [7,8], calculations of a -COOH group charge on the nanotubes surface, performed by the DFT method, showed that the charges on the atoms of the functional group were as follows: +0.4 on the carbon atom, (−0.3) and (−0.3) on oxygen atoms and +0.2 on the hydrogen atom. The formation of this group on the CNT boundary results in transferring electron density from the C atom of a -COOH group to the nanotube surface. In addition, the conductivity of the system increases. The carboxyl groups, showing an effect similar to nitrogen-containing ones, are the most active in oxygen reduction reactions (ORR) [7,8].

Atoms as nitrogen, phosphorus and sulphur are the most widely studied CM dopants [9–11]. Sizes of N and C atoms are comparable, making the doping process (car-

bon substitution) readily implemented [12]. The significant difference in electronegativity between N (3.0) and C (2.55) gives rise to the electron density shift from carbons to neighbouring nitrogen atoms. According to [13], the N atom in the carbon structure is highly negative (−0.277); however, it is compensated by three neighbouring carbon atoms with a positive charge. The presence of N atoms with a lone electron pair increases the electron-donating capability of CM. This ensures $O_2$ chemisorption in an orientation favourable for weakening (breaking) the O-O bond [14]. The neighbouring C and N atoms are characterised by Lewis basicity which makes them centres active to ORR [14,15]. In addition, nitrogen can take several configurations in the structure of CNT, as a result of the formation bonds with various atoms [12,16]. In this regard, pyridine nitrogen has one double and one single bond with carbon atoms in the benzene ring. Pyrrole nitrogen has two single bonds with C and with external hydrogen. Graphite nitrogen forms one double bond with carbon and two single bonds, one of them belongs to the neighbouring ring [12,16]. These differences in the configuration of the doping nitrogen provide its activity in interaction with oxygen. It was shown that pyridine N with a lone electron pair which does not participating in the bond formation with C atom plays an important role in increasing the CM activity in ORR [13]. According to the literature [17], depending on the nitrogen configuration in the carbon structure, both an increase and a decrease in the material electrical conductivity can occur. Graphite nitrogen is considered one of the most important active components because of its significant contribution to electrical conductivity [17]. The graphite nitrogen is bonded to three carbon atoms, one of which belongs to the neighbouring benzene ring. Moreover, three valence electrons of the N atom form σ-bonds, the fourth electron fills the p-state, and the fifth electron forms the π*-state, creating the p-doping effect that improves the conductivity of carbon materials. The pyridine nitrogen has a more complicated effect due to the distinct position of the N atom. It was reported [18] that pyridine nitrogen at the edges of the carbon matrix provides an additional electron in the delocalised π-system, which increases the conductivity of carbon materials. However, pyridine nitrogen in the basal planes generates many surface defects, leading to an increase in localised electronic states in the carbon matrix, which has negative effects on the electrochemical characteristics [19].

As a rule, oxygen-containing groups do not incorporate into the structure of a CM but bond to a carbon atom through a carbon atom. This is probably one reason for the less significant effect of oxygen-containing groups on the properties of the CNT than that of nitrogen [7,8].

Hydrophilic–hydrophobic properties and charge are important characteristics of the surface of a catalytic material [20]. The active centres on the surface of the N-doped CM increase its wettability, enhancing hydrophilic properties [19]. It was shown [21] that for nanoporous carbon doped with nitrogen, the adsorption of water molecules depends mainly on the quantity of nitrogen atoms, not on their configuration. According to a study [20], the mass activity of hydrophilic samples in ORR is much higher with onset potentials and half-wave shifting towards a positive direction. This indicates a high density of accessible active sites and their high dispersion owing to the highly hydrophilic surface of the CM. Consequently, using simultaneously hydrophilic materials, the amount of doping nitrogen and the value of a specific surface area can lead to the cumulative effect of increasing activity. However, the influence of surface hydrophilicity on the catalytic activity of the material has been rarely studied [20]. Thus, understanding the influence of surface hydrophilicity on electrocatalytic efficiency in ORR, calls for further research.

In this work, we modified CNT with O and N groups and further investigated the properties of obtained materials, such as electrical conductivity, hydrophilicity and zeta potential (ζ), which, in turn, determine their stability and catalytic activity in ORR.

## 2. Results and Discussion

The surface of the initial CNT is chemically inert [22,23]; therefore, to activate it, functionalisation and/or doping are necessary to form active centres. Functionalisation of

CNT with alkali yields a small number of oxygen-containing hydroxyl functional groups (532.3–532.8 eV) on the surface, as shown in Table 1.

**Table 1.** Structural characteristics of surface and pH of studied CNT.

| No. | Catalyst Treatment Conditions | Element/at.% | $S_{BET}$, m²/g | $S_{spec.}$, m²/g $C_8H_{18}//H_2O$ | $\zeta$-Potential in $H_2O$ | pH in Water from 6.5 in 30 min |
|---|---|---|---|---|---|---|
| 1 | $CNT_{NaOH}$ | O/2.18 | 297 | 333.5//49.2 | −1.47 | 8.2 |
| 2 | $CNT_{NaOH-N}$ | O/10.08 N/1.15 | 269 | 268.8//154.2 | −14.3 | 9.5 |
| 3 | $CNT_{HNO3}$ | O/15.4 N/1.14 | 300 | 342.6//145.6 | −41 | 5.7 |
| 4 | $CNT_{HNO3-N}$ | O/12.84 N/1.98 | 309 | 241.8//196 | −23.9 | 7.7 |

Upon treating with nitric acid, several types of functional groups are formed on the surface: ketone/carbonyl (531.0–531.9 eV), hydroxyl and carboxyl (532.3–532.8 eV) (Table 1). Furthermore, the total oxygen content on the surface increases from 2.18 at.% for CNT-1 to 15.43 at.% for CNT-3. Moreover, after treatment with nitric acid, the surface of CNT-3 contains 1.14 at.% of nitrogen, mainly in the pyrrole form (399.7–400.6 eV) (Table 1). As a result of nitrogen doping, mainly pyrrole (0.77 at.%) and pyridine (398.1–398.6 eV, 0.38 at.%) nitrogen forms occur on the CNT-2 surface (Figure 1). On the surface of CNT-4, 0.91 at.% pyrrole nitrogen and 0.54 at.% graphitized N (400.7 eV) are present. Figure 1 shows the surface composition of modified nanotubes analysed by X-ray photoelectron spectroscopy (XPS).

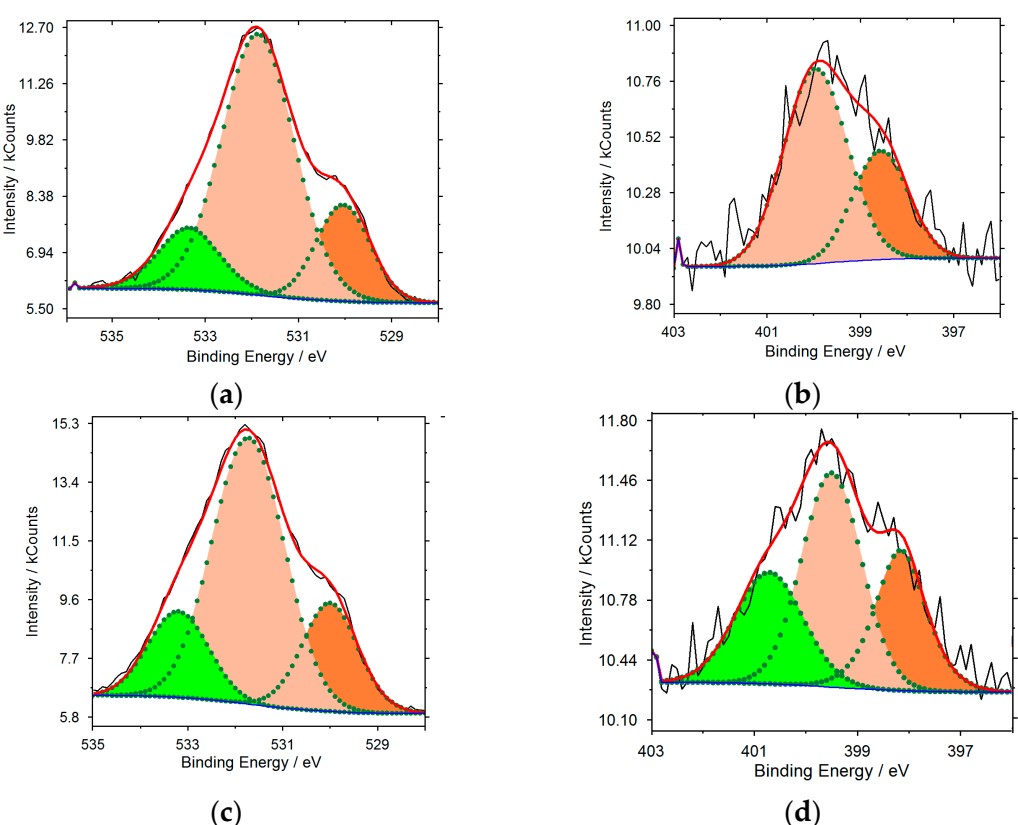

**Figure 1.** O1s (**a**,**c**) and N1s (**b**,**d**) X-ray spectra recorded on CNT-2 (**a**,**b**) and on CNT-4 (**c**,**d**).

For CNT-1 and CNT-3, the surface values determined by the BET and MSP using octane are similar are close. It should be noted that the hydrophilic surface is significantly larger in nanotubes treated with nitric acid. One possible reason is the large number of

different oxygen-containing groups in CNT-3. For in CNT-1, there are only OH$^-$ on the surface; while in CNT-3, there are mainly carboxyl groups, which cause an electron density shift similar to formation of pyridine nitrogen, its consistent with [7,8].

Possibly, this is due to the fact that carboxyl groups are the most active in ORR in alkaline media [24]. One can assume that a significant number of carboxyl groups on the surface, where hydrogen has a charge of +0.2 and carbon of +0.4 [7,8], leads to the adsorption of hydroxide ions. In this case, the pH of the CNT-3 solution shifts slightly towards lower values (Table 1). On the contrary, on the surface of CNT-1, there are only weakly bound groups OH$^-$ that can be desorbed and shift the pH of water towards higher values, as seen in the experiment. Accordingly, the zeta potential tends to zero. CNT-3 has the highest zeta potential value among the studied samples; this is associated with the adsorption of hydroxide ions from water. In alkaline solutions containing a large quantity of OH$^-$ ions, their adsorption on the negatively charged surface of carbon material is hindered; hence, oxygen is adsorbed and reduced on the electrode. This agrees with the fact that during desorption of hydroxide ions from the surface into the water, the zeta potential decreases (tends to zero in the case of CNT-1), and during the adsorption of OH$^-$ it increases (CNT-3). As a result of the modification, the electron density and electron-donor properties increase on CNT-3.

The change in the hydrophobic–hydrophilic properties is associated with CNT functionalisation and doping and is characterised by the amount of oxygen- and nitrogen-containing groups (Table 1). For CNT functionalised with alkali and acid, surface composition differs significantly. CNT-1 is characterised by the lowest hydrophilic surface area, compared with other samples, that comprises <15% of $S_{sp}$ measured by octane. CNT doped with nitrogen have the largest hydrophilic surface and the higher content of N atoms on the surface, the larger hydrophilic surface area (81% of $S_{sp}$ for CNT-4). Figure 2 shows the pore volume as a function of the logarithm of their radius when measured using octane and water. For all studied CNT, the water curve is shifted towards larger radii than the octane curve. This indicates a weaker wettability with water than with octane [25]; i.e., the average value of a wetting angle between water and all CNT samples exceeds zero. Measured in water and compared in the micro and mesopore region, the curves show that doping with N atoms results in an increase in the volume of hydrophilic pores with a radius of over 100 nm (Figure 2, Curves 2 and 4). Obtained for octane, the pore volume of these nanotubes, as well as the surface composition, is similar. This indicates that introducing N into the CNT structure leads to a decrease in the pore volume (regardless of the functionalisation type) and an increase in the proportion of hydrophilic pores. The functionalisation affects hydrophilicity only in the case of pores with a larger radius (Figure 2, Curves 1, 3 and 1′, 3′).

According to the data in Table 1 and Figure 2, the growth of oxygen-containing groups (mainly carboxyl) results in increase of the zeta potential and hydrophilic surface area. Upon subsequent nitrogen doping of functionalised CNT, the total pore volume measured with octane decreases, as well as the fraction of the volume of hydrophilic pores. For CNT-2 and CNT-4 the volume of hydrophilic pores is 1.2 cm$^3$/g, while the maximum volume of hydrophilic pores is 2.5 cm$^3$/g (CNT-3). The electrochemically active surface of CNT-3 is significantly higher than that of CNT-1 due to the large volume of hydrophilic pores (as will be shown further).

Electrical conductivity is one of the most important characteristics of electrode materials. The electrical conductivity of modified CNT is presented in Table 2.

CNT doped with nitrogen have the highest electrical conductivity. The high electrical conductivity for CNT-1 can be explained by lower (almost twice) material density than that of other CNT. Since the resistance was measured at a constant height between the electrodes (0.01 cm), the compaction of CNT with lower density is more prominent than that of CNT with high density. This leads to a higher number of contacts between neighbouring nanotubes and, consequently, increasing electrical conductivity. In the case of CNT-3, the low electrical conductivity (Table 2) is probably associated with the surface defects formed

during harsh oxidation and lower compaction than that of CNT-1. For CNT-2 and CNT-4, the electrical conductivity amounts to 0.283 and 0.212 S/cm, respectively.

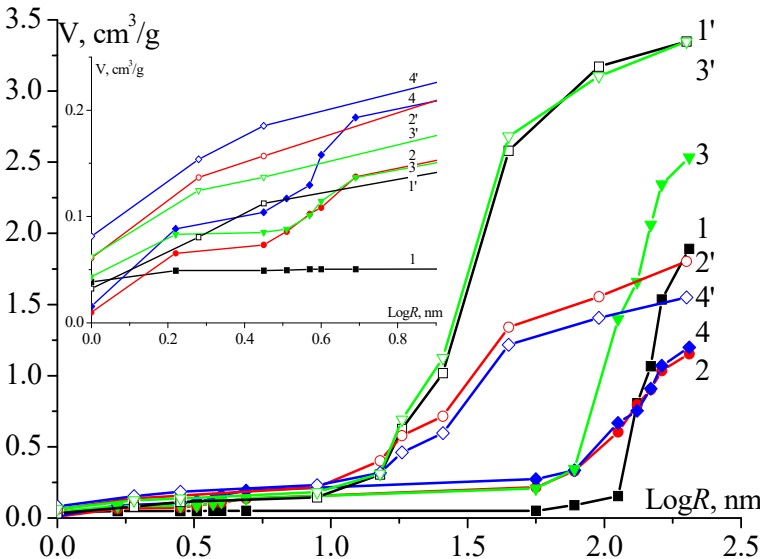

**Figure 2.** Integral curves of specific pore volume distribution *V* with respect to effective radii *R* (porometric curves) obtained with water (1, 2, 3, 4) and octane (1′, 2′, 3′, 4′) for CNT-1(1, 1′), CNT-2(2, 2′), CNT-3(3, 3′), and CNT-4 (4, 4′). The insert in the figure corresponds to the porometric curves in the area of micropores.

**Table 2.** Electrical conductivity and electrochemical characteristics of modified CNT.

| No. | Catalyst Treatment Conditions | $1/2Q$, C/g | $E_{1/2}$, V $\frac{1}{2}i_{lim.theoretical}$ | $i_{kin}$, mA/cm$^2$ //E, V (+0.05 V from $E_{stat}$) | $n$, Number of Electrons | $i$, mA/cm$^2$ (at 0.2 V) | $\kappa_{spec}$, S/cm | $\rho$, g/cm$^3$ * |
|-----|-------------------------------|-------------|-----------------------------------------------|-------------------------------------------------------|--------------------------|---------------------------|----------------------|---------------------|
| | | | in 0.1M KOH | | | in 0.5M H$_2$SO$_4$ | | |
| 1 | CNT$_{NaOH}$ | 21.5 | 0.64 | 0.12//0.75 | 1.6 | 0.124 | 0.234 | 0.407 |
| 2 | CNT$_{NaOH-N}$ | 61.2 | 0.81 | 0.7//0.83 | 2.6 | 1.895 | 0.283 | 0.90 |
| 3 | CNT$_{HNO3}$ | 79.5 | 0.71 | 0.15//0.81 | 1.8 | 0.450 | 0.17 | 0.58 |
| 4 | CNT$_{HNO3-N}$ | 62.0 | 0.82 | 0.9//0.83 | 3 | 2.329 | 0.212 | 0.85 |

* The density of modified CNT was calculated based on the data obtained by the MSP.

The Figure 3a,b shows CV of the studied CNT. The values of $S_{EAS}$ are given in Table 2. The electrochemical activity of modified CNT in ORR was investigated using 0.1 M KOH and 0.5 M H$_2$SO$_4$ electrolytes (Figure 3c,d). According to the polarisation curves the studied CNT exhibit high catalytic activity in an alkaline electrolyte.

The shape of the obtained curves suggests the reaction mechanism. In the acidic electrolyte, functionalised CNT-1 and CNT-3 are not active in ORR since the entire surface of negatively charged carbon material is filled with hydrogen adsorbed from the solution. The activity of nitrogen-doped CNT in ORR is slightly higher. The presence of nitrogen leads to a decrease in the volume and surface of pores, moreover significant part of which is hydrophobic and provide oxygen supply to the active centres. The polarisation capacity of this type of CNT is higher than that of CNT-1 (Figure 3). Due to the high electrical conductivity of CNT-2 and CNT-4, the activity in ORR in 0.5 M H$_2$SO$_4$ is higher than that on CNT-1 and CNT-3. In addition, the polarisation curve is positively shifted by ~0.300 V. The most probable factors influencing the activity are the low $S_{EAS}$ value of CNT-1 and the actual electrical conductivity; CNT-3 shows the lowest electrical conductivity among the studied samples. The overall reason for the low activity of CNT in an acidic electrolyte comprises blocking the surface for oxygen adsorption by adsorbed hydrogen. Modified CNT, primarily doped with nitrogen, exhibit high activity in an alkaline electrolyte. A high

negative value of the zeta potential and a shift in pH towards lower values (Table 1) can be explained by the adsorption of OH⁻ ions on the surface and are observed only for CNT-3. In this regard, the high activity of CNT-3 in the ORR stems from the higher $S_{EAS}$ value than that of CNT-1 (Figure 3). In an alkaline electrolyte, the entire $S_{EAS}$ remain accessible. OH⁻ ions do not adsorb on a negatively charged surface and do not hinder the oxygen adsorption and subsequent reduction. Thus, CNT modified with oxygen- and nitrogen-containing groups are ORR catalysts in an alkaline medium. In an alkaline electrolyte, CNT-1 is characterised by the lowest catalytic activity in the ORR due to the low content of heteroatoms on the surface and a small hydrophilic surface area. An increase in the activity of CNT-3 with respect to CNT-1 is associated with an increase in the number of oxygen-containing groups on its surface and the presence of nitrogen-containing groups increasing the hydrophilic and electrochemically active surface area (Table 2), as well as a high negative value of surface zeta-potential. The nitrogen-doped CNT are characterised by the highest activity in ORR, due to the presence in the surface composition of carboxyl groups, as well as pyridine and pyrrole nitrogen groups, ensuring a large hydrophilic surface area. The half-wave potential values are 0.81V and 0.82V for CNT-2 and CNT-4, respectively, being close to the $E_{1/2}$ values for Pt-catalysts [26]. In the case of CNT doped with nitrogen atoms, the current increases in the kinetic potential range, namely, 0.7 and 0.9 mA/cm² at 0.83 V for CNT-2 and CNT-4, respectively (Table 2).

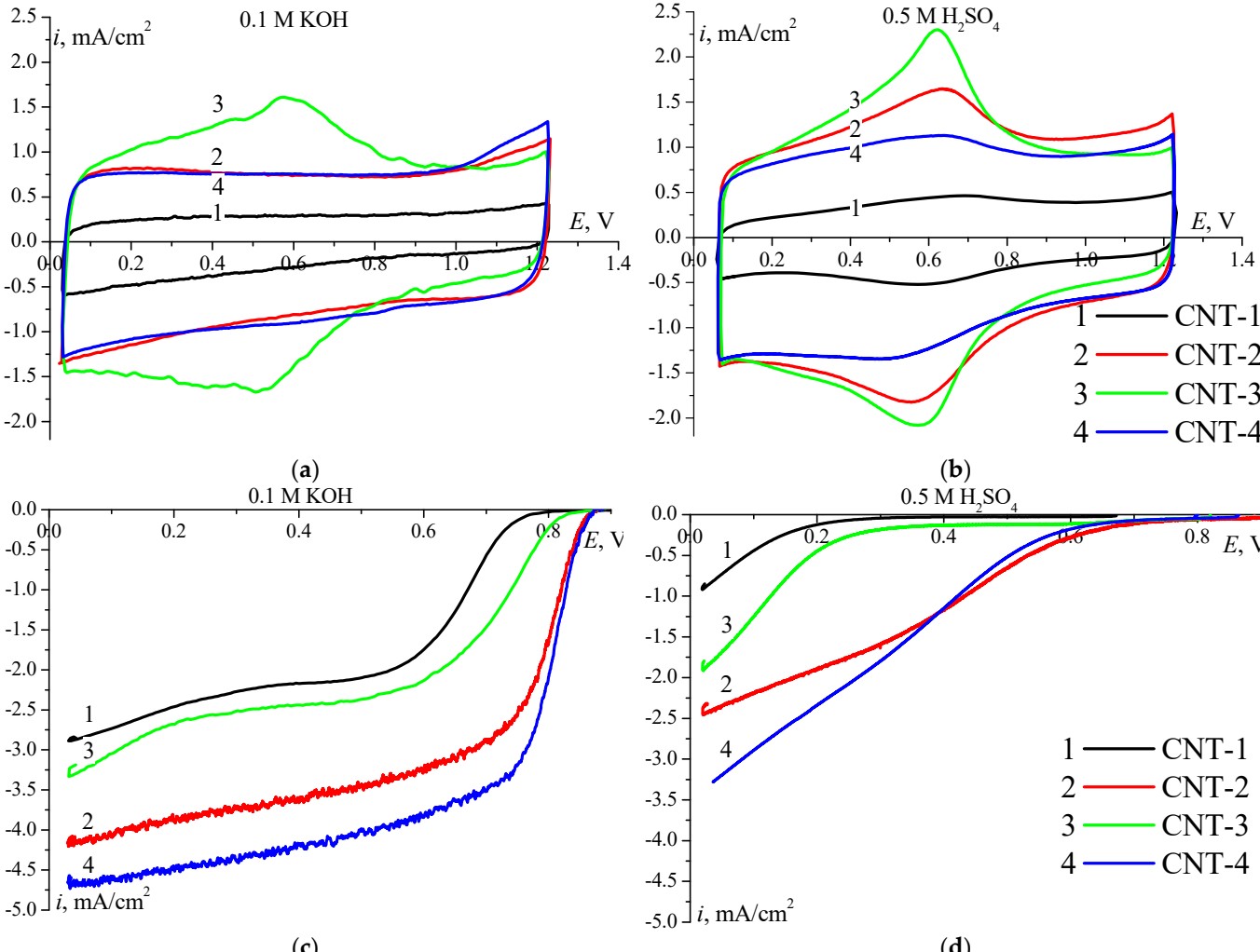

**Figure 3.** CV on modified CNT in 0.1 M KOH (**a**) and in 0.5 M H₂SO₄ (**b**) electrolyte solution. Ar-saturated, 100 mV/s, 0.150 mg/cm². Polarization curves of O₂ reduction on modified CNTs in 0.1 M KOH (**c**) and in 0.5 M H₂SO₄ (**d**) electrolyte solution, 5 mV/s, 1500 rpm, 0.150 mg/cm².

The corrosion stability of modified CNT was evaluated following the change in the $S_{EAS}$ value and the half-wave potential using the method of accelerated corrosion testing. All the studied materials are characterised by high stability within up to 1000 cycles (Figure 4). The maximum decrease in $S_{EAS}$ is observed at CNT-3 (13%) and may occur due to a large number of surface defects formed during the harsh oxidation with $HNO_3$. However, more surface defects promote the incorporation of more N atoms (Table 1), which significantly increase the CNT-4 stability. Thus, functionalising CNT with nitric acid increases its hydrophilic and electrochemically active surface area due to the formation of surface defects during harsh oxidation; however, its stability decreases compared to that of CNT-1.

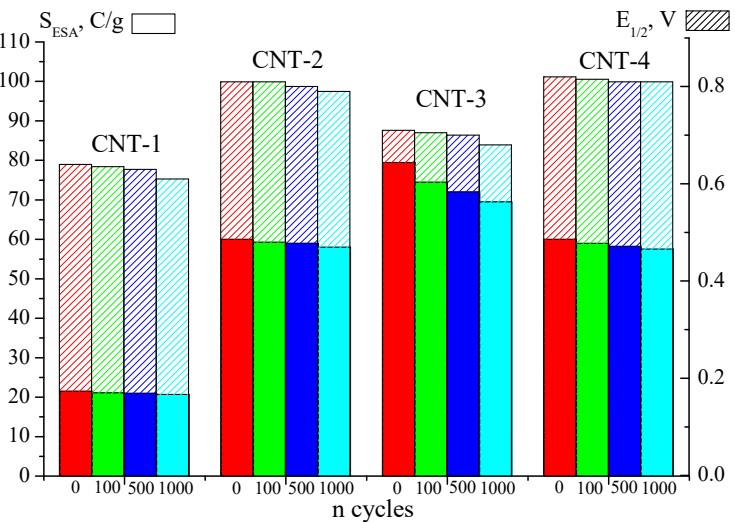

**Figure 4.** Variation of the $S_{EAS}$ and $E_{1/2}$ parameters of modified CNTs during accelerated corrosion testing in 0.1M KOH.

### 3. Materials and Methods

#### 3.1. Chemicals and Materials

Multiwalled CNT were supplied from Nanotechcenter LLC (Tambov, Russian) (>99.0% wt., $S_{BET}$ > 270 $m^2/g$, $d_{ext}$ = 10–30 nm). Melamine ($C_3H_6N_6$, >99.0%) were purchased from Alfa Aesar (Ward Hill, MA, USA). Potassium hydroxide—(KOH, 99.0%), sodium hydroxide—(NaOH, 99.0%), concentrated sulfuric—($H_2SO_4$, 95% wt.), nitric acid—($HNO_3$, 70% wt.) were purchased from Chimmed LLC (Moscow, Russia).

#### 3.2. Electrochemical Methods

Electrochemical measurements were carried out in a three-electrode electrochemical cell with separated electrode spaces. The working electrode was a glassy carbon electrode (GCE) (0.126 $cm^2$) pressed into a Teflon. A platinum wire was used as a counter electrode, the reference electrode was Hg/HgO electrode in alkaline (0.1 M KOH), and Ag/AgCl electrode in acidic (0.5 M $H_2SO_4$) electrolytes. All potentials are given relative to a reversible hydrogen electrode (RHE).

To prepare the catalyst ink, 2.2 µg of modified CNT was dispersed in 500 µL of isopropanol, 5 µL of this suspension spread on the GCE surface using a micro-syringe (~150 $µg_{cnt}/cm^2$). The catalytic ink was dried in air at room temperature for ~30 min.

Cyclic voltammetrywas used to determine $S_{EAS}$, which was assessed by the amount of electricity consumed to charge the surface of the studied materials (Q, C/g). CV were obtained at a 50 mV/s scan rate on a stationary electrode.

The polarization curves were used to determine the catalytic activity of the studied materials in ORR. The experiment was carried out in an $O_2$-saturated electrolyte, on a rotating disk electrode (650–3000 rpm), with a potential scan rate of 5 mV/s. The catalytic

activity was estimated on the half-wave potential ($E_{1/2}$, V), values of the limiting diffusion current density ($i_{lim}$, mA/cm$^2$), and current density in the kinetic region ($i_{kin}$, mA/cm$^2$).

To determine the corrosion resistance, the method of accelerated corrosion testing was used. The method consists in cycling the potential of the electrode in the range 0.6–1.3 V in 0.1 M KOH, at a potential scan rate 100 mV/s.After 100, 500, and 1000 cycles, S$_{EAS}$ was determined by CV in an Ar atmosphere and the activity in ORR was determined from polarization curves measured in a solution saturated with oxygen.

### 3.3. Resistance Measurement and Conductivity Calculation

In order to determine the electrical conductivity of synthesized materials, the resistance was measured using a measuring cell with the upper and lower disc-shaped contacts with an area of 0.785 cm$^2$, the latter was moved with a screw until it came into contact with a sample placed on the lower disk.

The resistance was measured by the electrochemical impedance method using a Solartron 1287 electrochemical interface. In the course of measuring the impedance spectrum, the dependence of the complex resistance on the frequency of the alternating current was obtained. The value of the experimental high-frequency resistance ($R_{ex}$) is equal to the value of the complex resistance, provided that it capacitive component ($Z^{//}$) is equal to zero.

Based on the experimentally measured $R_{ex}$ resistance, the specific resistance ($\rho_{spec}$) was calculated according to Equation (1):

$$\rho_{spec} = \frac{R_{ex} \times S}{h} \tag{1}$$

where $S$ = 0.785 cm$^2$, $h$ = 0.01 cm is the distance between the disks when they are compressed.

In turn, the $\kappa$ specific electrical conductivity [S/cm] was determined by Equation (2):

$$\kappa = \frac{1}{\rho_{spec}} \tag{2}$$

A more detailed description of the resistance measurement technique is given in [26].

### 3.4. Brunauer–Emmett–Teller (BET) Method

The specific surface area (S$_{BET}$) and porous structure of the studied materials were measured by the BET method using ASAP 2020 automatic analyzer setup (Micrometrics Instrument Corp., Norcross, GA, USA). Adsorption isotherms were obtained by physical sorption of nitrogen gas at 77 K.

### 3.5. X-ray Photoelectron Spectra (XPS)

XPS measurements were performed on an Auger spectrometer (Vacuum Generators, East Grinstead, UK) with the CLAM2 attachment, equipped with monochromatic Al K$\alpha$ radiation (200 W).The pressure in the analyzer chamber was maintained at a level of 10$^{-8}$ Torr. The data were corrected based on the position of a carbon C1s peak with an energy of 285.0 eV. For quantitative ratios, the sensitivity coefficients indicated in the VG1000 spectra processing program were used. The surface layer composition was determined up to 10 nm.

### 3.6. The Method of Standard Contact Porosimetry (MSCP)

To study the porous structure and hydrophilic–hydrophobic properties of CNTs, the method of Standard Contact Porosimetry was used [27,28]. This method can be used to study the porous structure of any materials in the widest possible range of pore radii from ~1 to 3 × 10$^5$ nm, and also their hydrophobic-hydrophilic properties. When octane is used as a measuring liquid, the obtained dependences ($V$-log $R$) include all the pores of the sample, when using water as a measuring liquid, only hydrophilic pores are measured.

### 3.7. Dynamic Light Scattering Method

The zeta potential of modified CNTs was determined by Zetasizer Nano ZSP, Malvern, using dynamic light scattering method. For measurements, suspension of CNTs (<1 µg) in water (2 mL) was prepared. The measurements were carried out at 25 °C using Clear disposable zeta cell [29].

### 3.8. pH Measurement

To measure the pH of aqueous suspensions, 10 mg of modified CNTs were added to 20 mL of deionized water (pH = 6.5) and periodically stirred. The measurement of pH was carried out using a high-precision pH-meter "Ecotest-120-pH-M", 30 min after the preparation of the suspension. An increase in the holding time did not lead to a change in pH.

### 3.9. Modification Methods

3.9.1. Functionalization of CNTs with Sodium Hydroxide Solution

A portion of the initial CNTs (400 mg) was placed into a flask with a backflow condenser and 200 mL of 1 M NaOH was added. The mixture was kept for an hour at a temperature of 100 °C with continuous stirring on a magnetic stirrer. Then the mixture was washed with deionized water to neutral pH and dried in a vacuum oven at 90 °C until the water was completely evaporated. CNTs after treatment with NaOH are called CNT-1.

3.9.2. Treatment of CNTs with Nitric Acid

Initial CNTs were placed in a concentrated $HNO_3$ solution. The mixture (500 mg CNT + 100 mL acid) was kept at 120 °C for 1 h. After cooling to room temperature, the mixture was washed with deionized water to neutral pH, and dried in a vacuum oven at 90 °C. CNTs after functionalization with nitric acid are denoted as CNT-3.

3.9.3. Nitrogen Doping

Functionalized CNTs was mixed with melamine ($C_3H_6N_6$) which used as a nitrogen precursor, in a ratio 1:0.7 and milled in a ball mill (Fritsch Pulverisette 7) for 1 h at 800 rpm. The resulting powder mixture was placed in a quartz tube and heat treated at 600 °C in an Ar atmosphere for 1 h. $CNT_{NaOH-N}$ doped with nitrogen are denoted as CNT-2, and $CNT_{HNO3-N}$—are denoted as CNT-4.

## 4. Conclusions

Modification of CNT with oxygen-containing groups has no effect on the specific surface area determined by BET and octane; however, it contributes to an increase in the hydrophilic surface area of the catalysts. To a greater extent, carboxyl groups on CNT-3 promote hydrophilisation. Subsequent doping with nitrogen leads to a decrease in the specific surface area, measured with octane, and an increase in hydrophilicity. The observed phenomena can be explained by a change in the electronic structure and surface charge due to the formation of active centres, i.e., O and N functional groups. The number and nature of functional groups affect the zeta potential value and the capacity for ion adsorption.

The zeta potential of CNT-1 is close to zero, as there is only weakly bound groups $OH^-$ on its surface, capable of desorbing from the surface to a solution with a neutral pH. A large number of carboxyl groups on the CNT-3 surface increase the electron density of the carbon material, leading to an increase in the zeta potential and an insignificant shift in the pH of the water to lower values. Doping with nitrogen, which incorporates into the CNT structure, leads to a further increase in the electron density of carbon material, thus preventing the adsorption of hydroxide ions. The observed changes in the zeta potential of modified CNT stem from the ratio of oxygen- and nitrogen-containing groups on their surface. Important information on the state of the surface of modified CNTs can be obtained by determining the dipole moments and work function. As shown in [30,31], these parameters change dramatically during functionalization. Determination of these

parameters is important for evaluating the efficiency of catalysts under electrocatalysis conditions. This direction is the subject of further research in relation to the reaction of oxygen electroreduction in alkaline electrolytes.

The high activity of modified CNT in ORR using alkaline electrolyte results from its high electron density. This provides adsorption of molecular oxygen in orientation favourable for breaking O-O bonds and effective oxygen reduction at overvoltage close to one observed on platinum catalysts [22]. The shape of the polarisation curves, high values of the half-wave potential and current density in the kinetic region indicate the electrocatalytic activity of modified CNT. CNT-2and CNT-4 have the highest activity in ORR in alkaline electrolytes owing to the presence of oxygen- and nitrogen-containing groups, which increase the electrical conductivity and enhance the electron-donating ability of carbon material.

Operating life (corrosion stability) is an essential characteristic of catalytic systems for their practical application. Express testing by the potential cycling method showed the following. Functionalisation with nitric acid leads to strong surface oxidation and the formation of a large number of oxygen-containing functional groups, partially replaced by nitrogen upon subsequent doping. Comparison of CNT-1 and CNT-3 showed less stability of the latter due to the formation of defects during harsh processing. Following nitrogen doping of functionalised CNT increases its corrosion stability, electrical conductivity and increases activity in ORR.

By the totality of characteristics, CNT modified by oxygen and nitrogen-containing groups are promising for further use as support in synthesis for metal-containing catalysts and/or developing active electrode layers in alkaline fuel cells.

**Author Contributions:** Conceptualization, supervision, V.B.; data curation, V.A.; investigation, methodology, I.V., M.R., V.S. and N.N. All authors have read and agreed to the published version of the manuscript.

**Funding:** This work was carried out with the financial support of the RFBR project BRICS_T No.19-53-80033.

**Data Availability Statement:** Additional data are available upon request from the corresponding author.

**Conflicts of Interest:** The authors declare no conflict of interest.

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
