# Peer review of "Modified Carbon Nanotubes: Surface Properties and Activity in Oxygen Reduction Reaction"

_catalysts, doi:10.3390/catal11111354_

Round 1

Reviewer 1 Report

In my opinion this is a good article, well designed and structured, which uses appropriate techniques to show the proposed objectives.

However, both the chemical modification of CNTs by O, N-functionalities is well known, as well as their influence on ORR processes. The authors previously present several articles with similar modifications and conclusions (eg Catalysts 10 (8), 1 - 10August 2020 Article number 892; Russian Journal of Electrochemistry, 2020, Vol. 56, No. 10, pp. 809–820). So the question to accept the article is to discern what is new, or what does this manuscript contribute with respect to the previous ones. Authors must specify this aspect, highlighting the progress made with respect to previous samples, or if it is a review of previous results.

Author Response

Reviewer 1.

  1. In my opinion this is a good article, well designed and structured, which uses appropriate techniques to show the proposed objectives.
  2. However, both the chemical modification of CNTs by O, N-functionalities is well known, as well as their influence on ORR processes. The authors previously present several articles with similar modifications and conclusions (eg Catalysts 10 (8), 1 - 10August 2020 Article number 892; Russian Journal of Electrochemistry, 2020, Vol. 56, No. 10, pp. 809–820).

Response: In addition to our articles, there is also a very wide range of publications devoted to the issues of modifying and doping CNTs by various methods and the effect of modifying on the activity of this type of carbon materials. Each of these works contributes to the development of ideas about changes in the composition and properties of the surface and their effect on the activity and stability of the carbon material under study. In these articles, other aspects are considered that affect the properties of the CNT surface for the implementation of the reaction of cathodic oxygen reduction.

  1. So the question to accept the article is to discern what is new, or what does this manuscript contribute with respect to the previous ones. Authors must specify this aspect, highlighting the progress made with respect to previous samples, or if it is a review of previous results.

Response: In answer to this question, we can say that this work presents new data characterizing the surface properties and structure of CNTs, such as the zeta potential and hydrophilicity, which vary depending on the nature (oxygen or nitrogen-containing) of the group modifying the surface. The new data obtained, taking into account the previously obtained results, allowed us to advance in understanding the laws that increase the activity and stability of modified CNTs as applied to the reaction of oxygen electroreduction in alkaline electrolytes. It is shown that oxygen-containing groups, as a rule, are bound to the CNT surface through a carbon atom and the influence of these groups affects an increase in the electrochemically active surface (polarization capacity). Nitrogen-containing groups are incorporated into the structure of CNTs, and their effect is more pronounced and is also expressed in an increase in electrocatalytic activity. The degree of change in properties is also determined by the number and nature of the modifying groups.

Reviewer 2 Report

The paper describes studies on the investigations on carbon nanotubes functionalized with oxygen and nitrogen-containing surface groups towards their application as materials for electrode layers. The paper is certainly interesting and topical and many reports can be found on the development of a robust electrodes for alkaline fuel cells this process. The work seems to be quite comprehensive, the results are reliable and I can recommend it for publication. However, I would like the Authors to interpret their results in terms of a molecular model. Nowadays we have tools to understand the role of surface functional groups on the most important aspects concerning electrode wettability and electrodonor properties. I would like the Authors to analyse the literature on the subject, where  more in-depth interpretation can be found see e.g. Covalently bonded surface functional groups on carbon nanotubes: from molecular modeling to practical applications, Nanoscale (2021); https://doi.org/10.1039/D0NR09057C

Or Insight into modification of electrodonor properties of multiwalled carbon nanotubes via oxygen plasma: Surface functionalization versus amorphization
Carbon 137 (2018) 425-432 doi.org/10.1016/j.carbon.2018.05.059

Based on the above reports, location of heteroatom (in-plane, aout of plane) type of functional group, surface coverage, topography  will play a fundamental role here.  Certainly microscopic observations are missing to know the actual  nature of the chemical treatment performed.

Final conclusion: minor revision

Author Response

The paper describes studies on the investigations on carbon nanotubes functionalized with oxygen and nitrogen-containing surface groups towards their application as materials for electrode layers. The paper is certainly interesting and topical and many reports can be found on the development of a robust electrodes for alkaline fuel cells this process. The work seems to be quite comprehensive, the results are reliable and I can recommend it for publication.

1.However, I would like the Authors to interpret their results in terms of a molecular model. Nowadays we have tools to understand the role of surface functional groups on the most important aspects concerning electrode wettability and electrodonor properties. I would like the Authors to analyse the literature on the subject, where  more in-depth interpretation can be found see e.g. Covalently bonded surface functional groups on carbon nanotubes: from molecular modeling to practical applications, Nanoscale (2021); https://doi.org/10.1039/D0NR09057C

Or Insight into modification of electrodonor properties of multiwalled carbon nanotubes via oxygen plasma: Surface functionalization versus amorphization
Carbon 137 (2018) 425-432 doi.org/10.1016/j.carbon.2018.05.059

Based on the above reports, location of heteroatom (in-plane, a out of plane) type of functional group, surface coverage, topography  will play a fundamental role here.  Certainly microscopic observations are missing to know the actual  nature of the chemical treatment performed.

Final conclusion: minor revision

Response: The authors thank the referee for analyzing our manuscript and providing links to publications in the field of research close to this work.

In these works, the analysis of the state of the surface of CNTs subjected to functionalization was carried out. The authors chose the work function and dipole moments as criteria for assessing the state of the surface. The results were obtained using DFT simulations. It was found that the dipole moments and work function change dramatically after modification. The degree of change depends on the nature of the groups (oxygen or nitrogen-containing) and their amount on the surface. We obtained similar conclusions in our work based on the results of measuring the zeta potential and surface hydrophilicity. It is suggested that nitrogen-containing groups localized in the structure of CNTs(in-plane) have a more effective effect on the surface properties, while oxygen-containing groups bound to the surface through a carbon atom (a out of plane) to a lesser extent affect the state of the surface..

In terms of further development of this area of ​​research, we propose carrying out calculations by the DFT method to optimize the process of modifying CNTs, which makes it possible to increase the activity of materials in the reaction of oxygen electroreduction.

Reviewer 3 Report

The article " Modified carbon nanotubes: surface properties and activity in oxygen reduction reaction" is devoted to the important problem of obtaining effective Pt-free catalysts of oxygen redaction reaction.

Nevertheless, there are a number of serious remarks to the article, presented below.

When studying the activity of materials, it is necessary to determine and compare the number of electrons (n) for oxygen reduction reaction.

In addition, it is important to value the amount of hydrogen peroxide released as an intermediate product for the materials studied.

Explain the peaks on CV (Figure 3 a, b) in the potential range of about 0.6 V for some samples. Why in Figure 3 b for material 3 for some reason the potential range is different

It is necessary to further clarify the synthesis conditions: why such a ratio of carbon - melamine, temperature and treatment time was chosen. Was this technique optimized or was the data taken from the literature.

More information is needed about carbon nanotubes, diameter, photos, why these nanotubes were chosen.

The designations of the catalysts are not very well thought out, for example, CNT-1 is used in the text, and in Tables 1 and 2 the material is designated as CNTNaOH, which leads to difficulties. Perhaps it is worth making a column in table 1 and 2 with the name "treatment conditions"

Why did you choose this particular method for assessing the stability of catalysts? Over 1000 CV for all materials, there was a slight degradation, which makes it difficult to choose the most stable material. How does this correlate with the data from the literature?

When presenting activity data, it would be useful to compare with commercially available platinum-free material or with literature data for similar materials. In addition, it would be useful to measure the activity of the nanotubes before treatment.

Author Response

1.When studying the activity of materials, it is necessary to determine and compare the number of electrons (n) for oxygen reduction reaction.

Response: A column corresponding to the number of electrons n in the oxygen reduction reaction has been added to table 2.

2 .In addition, it is important to value the amount of hydrogen peroxide released as an intermediate product for the materials studied.

Response: It is planned to carry out measurements by the method of rotating disk electrode with a ring (RRDE), which will allow us to correctly determine the value of n and the amount of hydrogen peroxide formed.

  1. Explain the peaks on CV (Figure 3 a, b) in the potential range of about 0.6 V for some samples. Why in Figure 3 b for material 3 for some reason the potential range is different

Response: Сharacteristic peaks at the potential of 0.6 V observed on the voltammetric curves of CNT-3 (Figure 3a,b) corresponds to the quinone-hydroquinone conversion. In the XPS spectrum of CNT-3, there are binding energy maxima at 531.5–531.9 eV, which can correspond to the C = O groups that are responsible for this conversion. On CNT-1, this maximum is weakly expressed, since only hydroxyl groups are present on the surface of these nanotubes, while quinone groups are absent and, therefore, there are no quinone / hydroquinone transformations.

 In Figure 3b, the CV curve for material CNT-3 is replaced by the CV curve recorded in the general potential range.

  1. It is necessary to further clarify the synthesis conditions: why such a ratio of carbon - melamine, temperature and treatment time was chosen. Was this technique optimized or was the data taken from the literature.

Response: Based on the literature data, we chose melamine and urea as precursors and developed a technique for doping CNTs with these materials. The most reproducible data were obtained using melamine at a CNT: melamine ratio of 1: 0.7. The method of functionalization and doping is described in the methodological part. Later this technique was used.

  1. More information is needed about carbon nanotubes, diameter, photos, why these nanotubes were chosen. Why did you choose this particular method for assessing the stability of catalysts? Over 1000 CV for all materials, there was a slight degradation, which makes it difficult to choose the most stable material. How does this correlate with the data from the literature.

Response: The choice of CNTs of this type is due to the results of previous studies and the information obtained on their structure and properties. The characteristics of the original CNTs are supplemented with pore sizes.

 The conditions and development of the method are the subject of our studies and the subsequent assessment of the efficiency of the synthesized catalyst in the oxygen reduction reaction.

Determination of stability by the potential cycling method is an express method for assessing the stability of catalysts and efficiency. This method depends on the measurement conditions: potential cycling interval, potential sweep rate and number of cycles. The conditions we have chosen are rather severe (the potential range is 0.6 - 1.3 V at a rate of 0.10 V / s). As follows from the experimental data, the most active decrease in the value of the electrochemically active surface is observed (activity) in the first 500 cycles, and then the surface (activity) remains practically unchanged, which gives us grounds to assess the stability of the systems under study based on the results of cycling (1000 cycles).

  1. The designations of the catalysts are not very well thought out, for example, CNT-1 is used in the text, and in Tables 1 and 2 the material is designated as CNTNaOH, which leads to difficulties. Perhaps it is worth making a column in table 1 and 2 with the name "treatment conditions"

Response: Added to table 1 and 2

  1. When presenting activity data, it would be useful to compare with commercially available platinum-free material or with literature data for similar materials. In addition, it would be useful to measure the activity of the nanotubes before treatment.

Response: In our previous work [27], we presented the results on the activity of the initial CNTs and their comparison with the data on the activity of a number of CNTs subjected to modification by various atoms. It was shown that modified CNTs significantly exceed the initial ones in activity in the reaction of oxygen reduction in an alkaline electrolyte. The most active of them (doped with nitrogen) are close to platinum in activity. In this regard, we do not present data for the initial nanotubes, since it is widely known from the literature that their activity is very low, and for those (before treatment) used in this work, this was shown in [27].

Round 2

Reviewer 3 Report

The authors answered the questions and made the necessary comments, so the article can be accepted for publication.